# SPARSE BINARY NEURAL NETWORKS

## ABSTRACT

Quantized neural networks are gaining popularity thanks to their ability to solve complex tasks with comparable accuracy as full-precision Deep Neural Networks (DNNs), while also reducing computational power and storage requirements and increasing the processing speed. These properties make them an attractive alternative for the development and deployment of DNN-based applications in Internet-Of-Things (IoT) devices. Among quantized networks, Binary Neural Networks (BNNs) have reported the largest speed-up. However, they suffer from a fixed and limited compression factor that may result insufficient for certain devices with very limited resources. In this work, we propose Sparse Binary Neural Networks, a novel model and training scheme that allows to introduce sparsity in BNNs by using positive 0/1 binary weights, instead of the -1/+1 weights used by state-of-the-art binary networks. As a result, our method is able to achieve a high compression factor and reduces the number of operations and parameters at inference time. We study the properties of our method through experiments on linear and convolutional networks over MNIST and CIFAR-10 datasets. Experiments confirm that SBNNs can achieve high compression rates and good generalization, while further reducing the operations of BNNs, making it a viable option for deploying DNNs in very cheap and low-cost IoT devices and sensors.

**Keywords :** Binary Neural Networks; Sparsity; Deep Neural Network Compression

## 1 INTRODUCTION

The term Internet-Of-Things (IoT) became notable in the late 2000s under the idea of enabling internet access to electrical and electronic devices (Miraz et al., 2015), thus allowing them to collect and exchange data. Since its introduction, the number of connected devices has managed to surpass the number of humans connected to the internet (Evans, 2011). The increasing number of both mobile and embedded IoT devices has led to a sensors-rich world, capable of addressing a various number of real-time applications, such as security systems, healthcare monitoring, environmental meters, factory automation, autonomous vehicles and many others, where both accuracy and time matter (Al-Fuqaha et al., 2015).

At the same time, Deep Neural Networks (DNNs) have reached and surpassed state-of-the-art results for multiple tasks involving images and video (Krizhevsky et al., 2012), speech (Hinton et al., 2012) or language processing (Collobert & Weston, 2008). Thanks to their ability to process large and complex multiple heterogeneous data and extract patterns needed to take autonomous decisions with high reliability (LeCun et al., 2015), DNNs have the potential of enabling a myriad of new IoT applications. DNNs, however, suffer from high resource consumption, in terms of required computational power, memory and energy consumption (Canziani et al., 2016). Instead, most IoT devices are characterized by their limited resources. They have limited processing power, small storage capabilities, they are not GPU-enabled and they are powered with batteries of limited capacity, which are expected to last over 10 years without being replaced or recharged (Global System for Mobile Communications, 2018). All these important constraints remain an important bottleneck towards deploying DNN models in IoT applications (Yao et al., 2018).

Achieving deployment of DNNs in IoT devices requires to compress deep neural networks to fit on IoT devices, while enabling real-time "intelligent" interactions with the environment (Yao et al., 2018) and without degrading their accuracy. Sparsity, compression and quantization, i.e. replacing

32 bit full-precision operations and values with fixed-point ones, can reduce inference time, storage size and power consumption of DNNs. This under the constraint of keeping high accuracy.

Different studies (Denil et al., 2013; Frankle & Carbin, 2019) have demonstrated that deep models contain optimal subnetworks, which can perform the same task of their related super-network with less memory and computational burden. Among the various techniques used to extract these subnetworks, pruning and quantization have shown promising results. The first one is based on removing parameters to obtain a sparser network, whereas the second focuses on reducing the bit-width to represent the parameters. Under this principle, Binary Neural Networks (BNNs) (Courbariaux et al., 2015) and Ternary Neural Networks (TNNs) (Hwang & Sung, 2014) are two recently proposed quantized neural networks with weights and activation functions using one and two bits, respectively. This approach avoids multiplication operations in the forward propagation, which are well-known to be computationally expensive, and replaces them with low-cost bitwise operations. This allows to speed-up the resulting networks and to compress them. For instance, BNNs with binary weights $\{-1, +1\}$ can reach a compression factor w.r.t. full-precision models of up to approximately 32 times (Rastegari et al., 2016). Despite this improvement the compression factor is upper bounded to 32, which is the result of representing the network's weights with 1-bit instead of the full-precision 32-bits, and may result insufficient for certain limited size and low power embedded devices.

To address this limitation, we introduce a novel quantized model denoted Sparse Binary Neural Network (SBNN). It shares the advantages of BNNs as it performs quantization using only one bit, while also introducing sparsity. Our SBNN uses $0s$ and $1s$ as weights, instead of $+1s$ and $-1s$ (Courbariaux et al., 2015; Rastegari et al., 2016), allowing to reduce the total number of required operations, and to achieve higher network compression rates and lower energy consumption at inference time. To achieve this, we propose a training scheme that starts from a "nearly-empty" model, rather than from fully connected models that prune their connections, as most state-of-the-art works do.

The remaining parts of this work is organized as follows. Section 2 discusses previous works to achieve sparsity and quantization in DNNs. The core of our contribution is described in Section 3. In Section 4, we study the properties of the proposed method and assess its performance, in terms of accuracy and compression results, through a set of experiments using MNIST and CIFAR-10 datasets. Finally, a discussion on the results and main conclusions are drawn in Section 5.

## 2 RELATED WORK

We review different approaches to address sparsity and quantization in quantized networks.

**Sparsity.** The concept of sparsity has been well studied beyond quantized neural networks as it reduces computational and storage requirements of the networks and it prevents overfitting. Methods to achieve sparsity either explicitly induce it during learning through regularization, such as $L_0$ (Louizos et al., 2018) or $L_1$ (Han et al., 2015) regularization; do it incrementally by gradually augmenting small networks (Bello, 1992); or by post hoc pruning (Srivastava et al., 2014; Srinivas et al., 2017; Gomez et al., 2019). In the context of quantized networks, Han et al. (2016) proposed magnitude-pruning of the nearly-zero parameters from the trained full-precision dense model followed by a quantization step of the remaining weights. The method achieved high compression rates of $\sim$35-49$\times$ and inference speed-up on well-known DNN topologies, without incurring in accuracy losses. However it has a time-consuming train-pruning stage and a relatively limited speed-up, which is model-dependent. Tung & Mori (2018) tried to optimize the scheme in Han et al. (2016) reporting an improvement in accuracy. However, their method encountered a smaller compression factor. Regarding TNNs, this type of networks naturally performs magnitude-based pruning thanks to their quantization function which maps real-valued weights to $\{-1, 0, +1\}$. Nevertheless, some works (Faraone et al., 2017; Marban et al., 2020) have achieved larger compression rates by explicitly inducing sparsity through regularization. Current BNN implementations do not address sparsity explicitly and focus on compression improvement through quantization (Rastegari et al., 2016). To account for sparsity, our work proposes to map real-valued weights to the positive $\{0, 1\}$ values, instead of the standard mapping of real weights to $\{-1, +1\}$.

**Quantization.** Network quantization allows the use of fixed-point arithmetic and a smaller bit-width to represent network parameters w.r.t the full-precision counterpart. As such, it has been

widely applied in graphs and neural networks since it reduces the power consumption and storage requirements of the network, while increasing the processing speed. Representing the values using only a finite set requires a quantization function that maps the original elements to the finite set. The quantization can be done either after training the model, using parameter sharing techniques (Han et al., 2016); or by directly quantizing the weights in the forward pass, as TNNs (Hwang & Sung, 2014), BNNs (Courbariaux et al., 2015) and other quantized networks (Hubara et al., 2017), in general. Similar to previous BNNs, our model also performs quantization during the training phase, while also introducing sparsity.

## 3 METHOD

We formulate the optimization problem addressed by SBNNs in section 3.1, then we present the algorithm to train our model and the proposed quantization functions (section 3.2). The sparse initialization is introduced in section 3.3. Finally, we describe implementation details and how SBNNs are more efficient in this aspect (section 3.4).

### 3.1 PROBLEM FORMULATION

Given a dataset of input-output pairs $\mathcal{D} = \{(x, y)\}$, the standard training of a full-precision neural network can be seen as a loss minimization problem:

$$\arg \min_{w} L(y, \hat{y}_w) \tag{1}$$

where $L(\cdot)$ is a loss function between true labels $y$ and the predicted values $\hat{y}_w$. The predicted values are a function of the data inputs and the networks' weights $w \in \mathbb{R}^N$, i.e. $\hat{y}_w = f(x; w)$.

Let us define a Sparse Binary Neural Network (SBNN) as a network with positive binary weights $w \in \{0, 1\}^N$ and a sparsity constraint imposing a maximum number $M$ of non-zero weights in the final network such that $\sum_i w_i \leq M < N$, with $N$ the total number of weights .The loss minimization problem from eq. (1) thus becomes a mixed discrete-continuous constrained optimization problem:

$$\begin{aligned}
\arg \min_{w} \quad & L(y, \hat{y}_w) \\
\text{s.t.} \quad & w_i \in \{0, 1\} \ \forall \ i, \\
& \sum_i w_i \leq M < N.
\end{aligned} \tag{2}$$

To simplify the mixed optimization problem, we first relax the discrete constraint imposed to the binary nature of the weights to deal with continuous variables. We thus allow for $w \in [0, 1]^N$ and we introduce a non-negative function $f_1(w)$, which penalizes values $w$ far from 0 and 1, while favouring the two extreme positive discrete values, i.e $f_1(0) = f_1(1) = 0$. Possible functions $f_1(w)$ are shown in fig. 1.

Furthermore, we relax the sparsity constraint, $\sum_i w_i \leq M$, by introducing a non-negative function $f_2(w)$ which penalizes non-zero weights. This can be achieved through $L_1$ or $L_2$ penalization. In this way, $f_1(w)$ pushes the real-valued weights towards 0 or 1, while $f_2(w)$ seeks to minimize the number of weights equal to 1. The effect of $f_1(w)$ and $f_2(w)$ is controlled using hyperparameters $\lambda_1$ and $\lambda_2$, respectively. Following (Murray & Ng, 2010; Srinivas et al., 2017), the relaxed optimization problem for a Sparse Binary Network can be expressed as:

$$\begin{aligned}
\arg \min_{w} \quad & L(y, \hat{y}_w) + \lambda_1 \sum_i f_1(w_i) + \lambda_2 \sum_i f_2(w_i) \\
\text{s.t.} \quad & w_i \in [0, 1] \ \forall \ i.
\end{aligned} \tag{3}$$

### 3.2 NETWORK TRAINING AND QUANTIZATION

The optimization problem from eq. (3) can be solved using the training algorithm proposed by Courbariaux et al. (2016) for standard BNNs with antipodal weights $w \in \{-1, +1\}$. Although

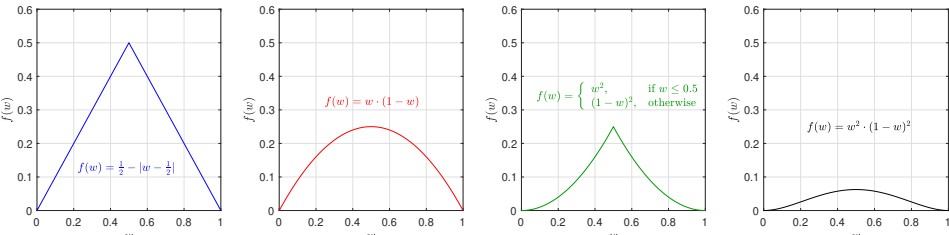

Figure 1: Possible regularization functions $f_1(\cdot)$. From left to right, triangular, $L_2$-inspired, piece-wise parabola and a $4^{th}$ degree polynomial.

the adaptation to positive weight constraints, i.e. $w \in \{0, 1\}$ is straightforward, the quantization function used to set the final weights to $\{0, 1\}$ requires reformulation. In the following paragraph, we first briefly describe the training procedure, followed by the formulation of the SBNN quantization function.

**Binary Network Training.**   The binary network training algorithm (Courbariaux et al., 2016) relies on Stochastic Gradient Descent (SGD) and it binarizes the network's weights during the forward propagation. To quantize the full-precision weights and back propagate their gradients, the sign function is used in the forward pass. Due to non-differentiability, the Straight-Through Estimator (STE) (Bengio et al., 2013) is used in the backward pass. During parameter update, no binarization is performed, as full-precision values are required for SGD to work. The magnitude of the real-valued weights beyond the binary values range does not influence the binarization operation in the forward pass. For this reason, after every parameter update those values are mapped to their nearest binary value. This is done through a clip operation in (Courbariaux et al., 2016). Finally, to reduce the effects of binary weight scaling, a batch-normalization layer (Ioffe & Szegedy, 2015) is always placed after a binary layer.

**Quantization Functions.**   As the optimization problem from eq. (3) allows for the networks weights to have continuous positive values in $[0, 1]$, we define a quantization function to map real-valued positive weights into binary positive values $\{0, 1\}$ during the training's forward pass and parameter update. Let us so define the weight quantization function $Q_w : [0, 1] \rightarrow \{0, 1\}$ as:

$$Q_w(x) = \begin{cases} 0, & \text{if } x \leq \frac{1}{2} \\ 1, & \text{otherwise} \end{cases} \tag{4}$$

Due to the introduction of positive weights, we modify the activation function used by each neuron, since networks with positive-valued weights in combination with the standard non-negative ReLU activation function, may lead to drop in the accuracy performance. Instead, the activation function used in our networks is the antipodal sign function, which can span both the negative and the positive realms, as in (Courbariaux et al., 2016). This function quantizes the outputs of each neuron in the antipodal binary domain $\{-1, +1\}$, and it is expressed by the quantization function $Q_a : \mathbb{R} \rightarrow \{-1, +1\}$ as:

$$Q_a(x) = \begin{cases} -1, & \text{if } x < 0 \\ +1, & \text{otherwise} \end{cases} \tag{5}$$

The activation function follows the batch-normalization layers.   The effect of the batch-normalization with its learnable parameters can be seen as the effect of moving the threshold $0$ (eq. (5)) of the activation function to a new threshold value $\gamma$ (Umuroglu et al., 2017). This effect is needed by the network to reduce the quantization error $\varepsilon = x - Q(x)$, i.e. the incurred error when mapping full-precision values to the quantized binary domain. Although this results in that batch-normalization introduces more parameters and extra-computations during training (e.g. mean and variance of inputs over the mini-batch), when the trained model is deployed, the new threshold $\gamma$ can be stored using only one integer parameter (Umuroglu et al., 2017), thus requiring only a comparison and no further computations.

### 3.3 SPARSE INITIALIZATION

Differently from standard network initialization, in SBNNs every weight follows a Bernoulli distribution of parameter $p$:

$$P(w_i = 1) \sim \mathcal{B}(p).$$

The sparsity parameter $p$ controls the number of initial connections in the network. A fully connected network is achieved with $p = 1$, whereas an initial empty network is obtained with $p = 0$. Our proposed training scheme works under the hypothesis that training is faster starting from a sparse initialization $p$ close to zero, and adds only the necessary connections, rather than starting from a fully connected network and removing most of its connections. At the same time, a too sparse initialization $p \approx 0$ may induce large gradient errors, forcing the network to add a great number of connections. A Bernoulli distribution to remove connections is exploited also in Ardakani et al. (2016), with the idea of creating a mask for binary and ternary weight networks that selects only a percentage $p$ of weight parameters to control the sparsity level. Their proposed approach is really effective using VLSI technology, but due to the transformation from binary weight networks with $w \in \{-1, +1\}$ to ternary values $w \in \{-1, 0, +1\}$, the sign of the non-zero connections needs to be encoded in normal processors, adding overheads in the encoder and decoder, while limiting the compression factor.

### 3.4 IMPLEMENTATION

BNNs replace the multiplication and accumulation operations in the forward pass of full-precision networks by two bitwise operations. Multiplication is replaced by a XNOR operation between binary weights at a given layer and the binary inputs from previous layers, and accumulation by a popcount operation.

Connections in the SBNN formulation will always have the same weight of 1. As such, XNOR operations are not required allowing SNNs to further simplify the computations of the network. The only operation needed by SBNNs during the forward pass is the popcount. Furthermore, it is performed only among the connected input bits through the $1s$ weights rather than the full input. The implementation gain of SBNNs w.r.t. BNNs is illustrated in fig. 2.

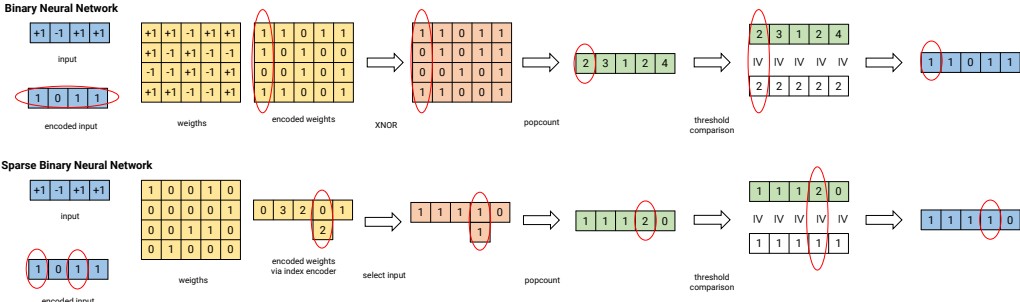

Figure 2: Example of implementation of Binary Neural Networks and Sparse Binary Neural Networks. The threshold in Binary Neural Networks is $\lceil \frac{n_{inputs}}{2} \rceil$, with $n_{inputs}$ the dimension of each input sample at a given layer. In Sparse Binary Neural Networks, this accounts to $\lceil \frac{n_{1s}}{2} \rceil$, with $n_{1s}$ the number of connected inputs at each output neuron.

## 4 EXPERIMENTS AND RESULTS

We trained and tested the proposed SBNN on classification problems from two different datasets widely used to understand the generalization of BNNs, using similar network topologies (Courbariaux et al., 2015; 2016; Hubara et al., 2017; Kim & Smaragdis, 2015): 1) the MNIST dataset (LeCun & Cortes, 2010), consisting of 28×28 black and white images of handwritten digits; and 2) CIFAR-10 (Krizhevsky et al.), consisting of 32×32 RGB images of natural scenes. Both datasets have 10K images for testing and 10 different classes, while they have 60K and 50K images for training, respectively.

In the following section we report experimental results obtained on each dataset, where we study the properties of the proposed SBNN. Concretely, we use MNIST to study SBNNs using linear layers, whereas CIFAR-10 is used to study the behaviour of SBNNs with convolutional layers.

## 4.1 MNIST

This section first describes the setup used to train our SBNNs over MNIST dataset. Then we study how the sparsity parameter $p$, the learning rate, and the use of regularizers $f_1$ and $f_2$ affect the SBNNs sparsity and generalization capacity. Lastly, we compare our SBNNs with state-of-the-art BNNs in terms of accuracy and compression.

**Setup.** We use the linear topology proposed by Courbariaux et al. (2016), consisting of 3 hidden layers with 1024 neurons each, followed by batch-normalization layers with learnable parameters and the sign function. We denote our sparse architecture as 3L-SBNN and provide further details about it and its implementation in appendix A.1.

**Effects of sparsity parameter $p$.** In this first experiment, we study the sparsity and the generalization properties of our SBNN as a function of the initial number of connections, controlled by $p$. Sparsity, quantified by the number of effective connections, EC $= |W \neq 0|/|W|$ (Marban et al., 2020), and generalization, measured in terms of test accuracy, over training time (epochs) for different values of $p$ are reported in fig. 3. The learning rate was fixed to 1. A sparse network will have an EC close to zero, whereas a fully connected network has EC $= 100\%$.

Results show that a very low number of initial connections (e.g. $p < 1\%$) seems to induce large gradient errors, leading to a large increment of connections, clearly visible after the first epoch. Conversely, larger values of $p$ induce smaller variations in the number of connections over time. Regarding training time, low values of $p$ (e.g. $p \leq 1\%$) allow for a faster training, which is manifested through the fact that these achieve smaller generalization errors faster than higher values of $p$ (fig. 3 right).

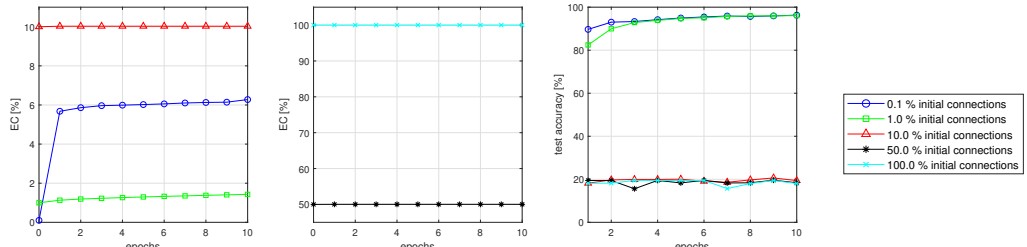

Figure 3: Effective Connections (EC) as a function of the initial number of connections ($p$) in a Sparse Binary Neural Network with 3 feed-forward hidden linear layers with 1024 neurons each, trained on MNIST dataset. **Left.** $p = 0.1\%$ (O), $p = 1.0\%$ ($\square$), $p = 10.0\%$ ($\triangle$). **Center.** $p = 50.0\%$ (*) and $p = 100.0\%$ ($\times$). **Right.** Corresponding test accuracy.

**Effects of the learning rate.** Here we study the sparsity and generalization of SBNNs as function of the learning rate. Using a fixed $p = 1.0\%$, which resulted in previous experiment in a good trade-off between sparsity and generalization, fig. 4 reports how the number of connections varies over training time along with test accuracy.

As it would be expected results show that higher learning rates lead to a faster convergence, manifested by a faster increase in test accuracy. Regarding the number of network connections, however, smaller learning rates favour sparsity: the smaller the learning rate, the smaller the increase in the number of connections over time. Large learning rates ($lr = 10$) are catastrophic for sparsity (fig. 4 left). Overall, results indicate it is possible to find a compromise between learning rates that can achieve high accuracy within reasonable training times, while preserving sparsity.

**Effects of the regularizers.** In this last experiment, we study the role of the regularizers (eq. (3)) in the SBNN's performance in terms of sparsity and accuracy. We compare EC and test accuracy

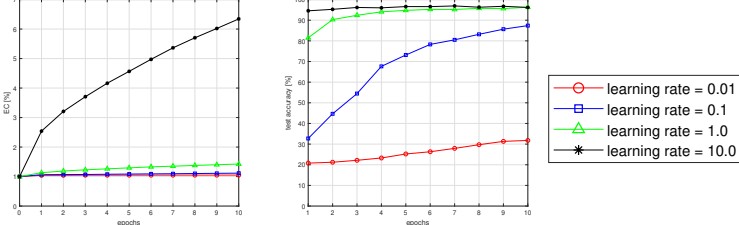

Figure 4: **Left.** Effective Connections (EC) as a function of the learning rate ($lr$) in a Sparse Binary Neural Network with 3 feed-forward hidden linear layers with 1024 neurons each, trained on MNIST dataset. Each curve represents a different learning rate, $lr = 0.01$ (O), $lr = 0.1$ ($\square$), $lr = 1.0$ ($\triangle$), and $lr$=10.0 ($*$). **Right.** Corresponding test accuracy.

Table 1: Effects of $f_1$ and $f_2$ in test accuracy (ACC) and effective connections (EC).

| $f_1$ | $f_2$ | | | |
|---|---|---|---|---|
| | $L_2$ | | $L_1$ | |
| | ACC [%] | EC [%] | ACC [%] | EC [%] |
| triangular | 97.48($\pm$0.05) | 1.27($\pm$0.07) | 96.96($\pm$0.12) | 1.21($\pm$0.01) |
| L2-inspired | 97.38($\pm$0.16) | 1.17($\pm$0.03) | 97.31($\pm$0.09) | 1.01($\pm$0.03) |
| piecewise-parabola | 96.99($\pm$0.10) | 1.33($\pm$0.02) | 97.27($\pm$0.12) | 1.17($\pm$0.03) |
| $4^{th}$ degree polynomial | 97.51($\pm$0.02) | 1.44($\pm$0.20) | 97.69($\pm$0.08) | 1.66($\pm$0.03) |
| no regularizers | ACC [%]: | 97.78($\pm$0.07) | EC [%] : | 2.03($\pm$0.03) |

results with no regularization, versus the case of using different $f_1$ regularizers: triangular, $L_2$-inspired, piece-wise parabola and a $4^{th}$ degree polynomial (fig. 1); and an $L_1$ and $L_2$ penalization for $f_2$. The models are trained for 40 epochs, with initial learning rate of 2, decreased by 10 times every 15 epochs. $\lambda_1 = \lambda_2 = 1e{-}5$, except for the triangular function where $\lambda_1 = 1e{-}6$.

Results in table 1 show that regularizers achieve higher sparsity levels (less final connections), although they tend to have a slight drop in accuracy. The latter could be explained by the fact that the minimization problem does not involve only the loss function, but a sum of the loss with $f_1$ and $f_2$. Increasing the learning rate or the number of epochs should compensate for the accuracy loss.

**Method Comparison.** Finally, we compare the accuracy, effective connections (EC) and compression performance of the proposed SBNN with state-of-the-art BNN formulations. The compression performance is measured in terms of the compression rate, which is computed as the total number of bits to represent each network parameter at full-precision, over the number of bits of full-precision parameters and binary parameters of the network. We use, for this purpose, three different encoders: the index encoder (IE), run-length encoder (RLE) and huffman encoder (HE); and we compare them with the case with no encoder (NE). Details on the compression procedure are found in appendix B.

We assess the SBNNs using two topologies: the 3L-SBNN and a 2 hidden layers variant of it, 2L-SBNN. We compare our results with the following BNNs: 1) 3L-BNN, the 3 layer BNN from Courbariaux et al. (2016); 2) 2L-BNN, the 2 layer variant of 3L-BNN; and 3) Bitwise (Kim & Smaragdis, 2015). The networks are trained for 40 epochs, starting with a learning rate = 2 and decreasing it by a factor of 10 every 15 epochs. For SBNNs, $p = 1\%$ and $\lambda_1$ and $\lambda_2$ are set to 0. As we could not reproduce the results of Bitwise, we report those in Kim & Smaragdis (2015).

The results obtained for the different models are reported in table 2. As it can be seen, SBNNs achieve the best compression rates among all methods both with 3L-SBNN and 2L-SBNN topology, incurring in a loss of less than $1\%$ w.r.t Bitwise, which is the Binary Neural Network reporting the highest accuracy. In particular, SBNNs outperform other BNNs by over 4 times in compression when using a Huffman encoder. This result is of high practical interest. It suggests that SBNNs for simple tasks can be implemented in very cheap and severely memory limited IoT devices, ensuring at the same time low consumption and fast inference, therefore opening the possibility to automate various simple tasks at low cost. For instance, the process of recognizing handwritten ZIP codes

Table 2: Method comparison on MNIST dataset. Values in parentheses report standard deviation.

| Model | ACC [%] | Compression rate | | | | EC [%] |
|---|---|---|---|---|---|---|
| | | NE | IE | RLE | HE | |
| 2L-SBNN **[our]** | 97.83($\pm$0.07) | 32 | 101 | 111 | 131 | 2.65($\pm$0.03) |
| 2L-BNN | 98.30($\pm$0.08) | 32 | - | - | - | 100 |
| Full-precision | 98.72($\pm$0.07) | 1 | - | - | - | 100 |
| 3L-SBNN **[our]** | 97.78($\pm$0.07) | 32 | 128 | 144 | 173 | 2.03($\pm$0.03) |
| 3L-BNN | 98.15($\pm$0.07) | 32 | - | - | - | 100 |
| 3L-Bitwise | 98.67 | 32 | - | - | - | 100 |
| Full-precision | 98.65($\pm$0.04) | 1 | - | - | - | 100 |

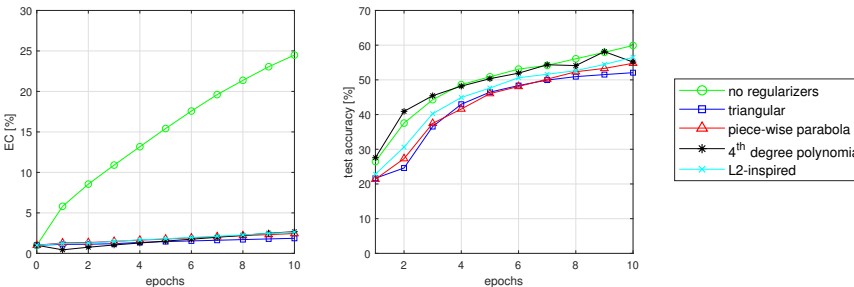

Figure 5: Effects of $f_1$ and $f_2$ in sparsity and test accuracy of a SBNN with convolutional topology trained on CIFAR-10. **Left.** Effective Connections (EC). **Right.** Corresponding test accuracy.

in post offices or measurement reading from screens of non internet-enabled meters sending them through an IoT network to a central office to process for rapid anomaly detection and localization.

In the final experiment, we make a comparison between SBNNs and 3L-BNN with dropout (Srivastava et al., 2014) as this combination could lead to a final sparser BNN. Over 40 training epochs on MNIST, SBNN outperforms 3L-BNN with dropout by a large margin both in sparsity and accuracy. For this reason, the results are not included but, for the sake of completeness, we report them in appendix C.

## 4.2 CIFAR-10

This section first describes the setup used to train our convolutional SBNNs over the CIFAR-10 dataset. Then we propose an experiment to study the relation between sparsity of convolutional SBNNs and the use of regularizers $f_1$ and $f_2$. Lastly, we compare our SBNNs with state-of-the-art BNNs in terms of accuracy and compression.

**Setup.** We use a convolutional topology inspired on the VGG network (Simonyan & Zisserman, 2015). We use an $L_2$-inspired for $f_1$, and the $L_2$ norm for $f_2$ regularizers, with hyperparameters $\lambda_1 = 7e-6$ and $\lambda_2 = 3e-6$. The training set was augmented through the use of different random rotations, crop and flips at every epoch and for each image. A detailed description of the network architecture and the training procedure is provided in appendix A.2.

**Effects of the regularizers.** We study the sparsity and the generalization properties of SBNNs on convolutional networks, when regularizers are used. We assess sparsity with EC and generalization in terms of test accuracy, over training time (epochs) for the different $f_1$ (see fig. 1) and $f_2$ ($L_1$ and $L_2$) regularizers . Results in fig. 5 are obtained over 10 training epochs with a learning rate of 0.01. As Courbariaux et al. (2015), we use an Adamax optimizer, since it shows faster generalization of BNNs. The use of regularizers in convolutional SBNNs, as with linear topologies, leads to higher sparsity levels at the cost of a slower convergence. However, differently from linear networks, their presence is clearly needed to preserve sparsity.

Table 3: Method comparison on CIFAR-10 dataset. Values in parentheses report standard deviation.

| Model | ACC [%] | Compression rate | | | | EC [%] |
|---|---|---|---|---|---|---|
| | | NE | IE | RLE | HE | |
| SBNN [our] | 78.59($\pm$0.47) | 32 | 108 | 119 | 145 | 3.10($\pm$0.01) |
| BNN | 88.18($\pm$0.33) | 32 | - | - | - | 100 |
| Full-precision | 88.94($\pm$0.77) | 1 | - | - | - | 100 |

**Method Comparison.** We compare the proposed SBNN with the state-of-the-art BNN formulations proposed in (Courbariaux et al., 2016) in terms of accuracy, effective connections (EC) and compression performance. We use the same compression techniques and performance measurements used for MNIST. Both networks are trained for 300 epochs, starting with a learning rate of 0.05, decreased by a factor 2 firstly after 100 epochs, and then every 50 epochs. For SBNNs, $p$ is set to 1.0%, while $\lambda_1$ and $\lambda_2$ are set to $7\mathrm{e}-6$ and $3\mathrm{e}-6$, respectively. The results for the different models reported in table 3 confirm the SBNNs superiority in terms of EC and compression rates. However, differently from the linear topology, the convolutional SBNN incurs in an higher accuracy loss of $\sim 10\%$ w.r.t. BNNs.

## 5 Conclusions

We propose Sparse Binary Neural Network (SBNN), a method to further compress Binary Neural Networks (BNNs) by introducing sparsity, while reducing their required computations. Our approach is based on quantization of weights in the 0/1 binary domain and a highly sparse initialization of the network. It is formulated as a mixed optimization problem and solved using a modified version of the BNN training algorithm with -1/+1 weights (Courbariaux et al., 2016). The method has been evaluated on feed-forward linear and convolutional network on MNIST and CIFAR-10 data sets, respectively. The achievable compression rate of SBNN is much higher than simple BNNs, making it a feasible alternative for IoT devices and sensors. For instance, 2L-SBNN with a model size of 70 kB (with index encoder) and 97.83 % accuracy on MNIST, can be entirely stored, not only in the flash memory, but also in the SRAM memory of very low-power hardware modules like the Intel® Curie™ Module, which has 384kB of flash memory and only 80kB of SRAM.

A current weakness of SBNNs is that sparsity, speed-up and high compression rates come at the cost of reduced performance accuracy. While the generalization error in linear architectures remains competitive, this has a larger drop in convolutional networks. The problem of generalization is not exclusive of SBNNs, and it is common also to +1/-1 BNNs. As discussed in Qin et al. (2020), it is still unclear what kind of network structure is suitable for binarization, so that the information passing through the network can be preserved, even after the binarization itself. As a result, multiple methods have been proposed to mitigate generalization losses by reducing the gradient (Rastegari et al., 2016; Yin et al., 2018; Darabi et al., 2019) and the quantization error (Kim & Smaragdis, 2015; Liu et al., 2018). Future works should continue in this direction with a specific focus on the 0/1 binary domain. Another interesting direction could be to extend the learning capabilities of SBNNs by investigating how to extract them from sparse ternary networks.

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

APPENDIX

## A  LINEAR AND CONVOLUTIONAL TOPOLOGIES SETUPS

This section provides details about the network configurations used in the experiments. We provide details on the linear architecture used for experiments with the MNIST dataset and the convolutional topology used with CIFAR-10.

### A.1  LINEAR ARCHITECTURE

For experiments with MNIST, we use the linear topology proposed by Courbariaux et al. (2016). The network consists of 3 hidden layers with 1024 neurons each, followed by batch-normalization layers with learnable parameters and the sign function. The LogSoftMax activation function replaces the sign after the output layer, which contains 10 neurons. The loss used is the Negative Log-Likelihood and the optimizer is Stochastic Gradient Descent (SGD) with Nesterov momentum $\mu = 0.9$. We denote this architecture 3L-SBNN (i.e. SBNN with 3 hidden linear layers). To speed up the training we used a mini-batch of 200 samples.

### A.2  CONVOLUTIONAL NEURAL NETWORK

For the experiments with CIFAR-10, we use a convolutional topology inspired on the VGG network (Simonyan & Zisserman, 2015). The 32×32 RGB images (3 channels) pass through 6 convolutional layers with 3×3 kernels, stride = 1 and zero padding = 1. Each layer is followed by a batch-normalization layer and the sign activation function. Every 2 layers, we add a max-pooling layer with a 2×2 kernel and stride = 2, before the batch-normalization layer. The first 2 convolutional layers have 128 channels, the following 2 have 256 channels, while the last ones 512 channels. The output of the last convolutional layer is flattened and followed by a linear layer with 10 neurons (one per class) followed by a batch-normalization layer and LogSoftMax activation function. Similarly to MNIST, the loss used is the Negative Log-Likelihood. We use Adamax as optimizer and a scheduler which halves the learning rate after 100 epochs and then halves it every 50 epochs. The models are trained for 300 epochs in total. As regularizers, we use an $L_2$-inspired for $f_1$, and the $L_2$ norm for $f_2$. The hyperparameters $\lambda_1$ and $\lambda_2$ are set to 7e−6 and 3e−6, respectively. To speed up the training, we used a mini-batch of 200 samples.

Finally, to enlarge the training set, this was augmented through the use of different random rotations, crop and flips at every epoch and for each image.

## B  COMPRESSION TECHNIQUES

Well-known methods from source coding theory can be exploited for the compression of sparse vectors and matrices. An SBNN can be seen as a set of sparse binary matrices. To evaluate the compression potential of SBNNs, we have used three different encoders in our experiments: index encoder, run-length and Huffman.

An illustration of the index encoder and the run-length encoders is depicted in fig. 6. The first encodes only the column (or row) indexes of the $1s$ elements, the latter encodes the length of the run-length of zeros. Since the distribution of the run-lengths of zeros in the matrices of SBNNs varies, the length in bits to represent each run-length, as shown in fig. 6, is chosen with a full search of all possible lengths in bits from 1 to $\log_2$(maximum run-length).

The third approach, the Huffman encoder (Huffman, 1952), maps elements, in our case indexes of $1s$ elements, to bit-symbols of variable length, depending on the frequency of each element: more frequent elements are encoded using less amount of bits, while less frequent elements are mapped to larger in bit symbols.

The reported compression rates (table 2 and table 3) are obtained as a ratio of the equivalent full model size over the SBNN one. The full model size is computed considering each weight stored with 32-bit plus 32-bits for each scaling factor introduced by batch-normalization layers. The SBNN

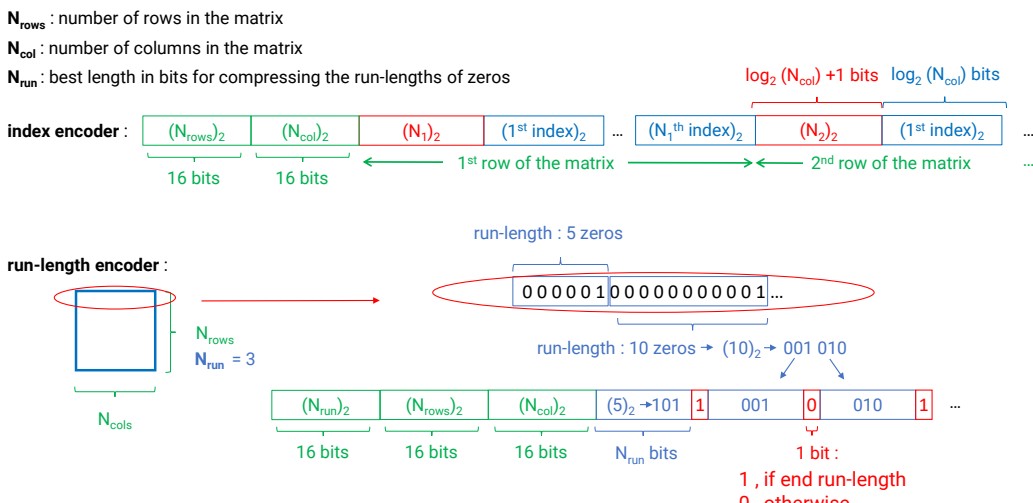

Figure 6: Schematics of the index encoder and of the run-length encoder used to compress the sparse binary weigth matrices

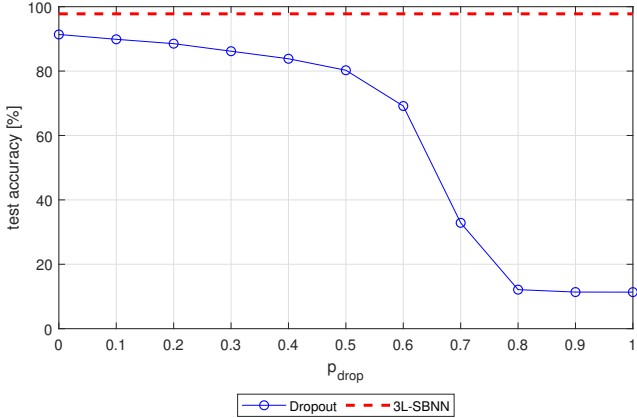

Figure 7: The blue curve shows the test accuracy as a function of dropout probability ($p_{\text{drop}}$) in the 3L-BNN with dropout, trained on MNIST dataset. The red curve represents the accuracy of a Sparse Binary Neural Network with same topology, no dropout.

size is computed compressing the binary matrices with the three different encoders, while the integer thresholds $\gamma$ of the sign activation function are stored with 16 bits each.

## C  DROPOUT

This section provides further details on the comparison of the 3L-SBNN with a 3L-BNN in combination of dropout (Srivastava et al., 2014) for different values of $p_{\text{drop}}$, the dropout probability for each neuron. Larger $p_{\text{drop}}$ means less neurons. A $p_{\text{drop}}$ of 0 corresponds to nearly a full-connected network, while a $p_{\text{drop}}$ of 1 to a nearly empty network. For consistency, the 3L-BNN with dropout and SBNN are trained for 40 epochs with learning rate = 1.0, and decreasing it of 10 times every 15 epochs. No regularizers were used for the SBNN. We report the results, in terms of accuracy for different values of $p_{\text{drop}}$, in fig. 7.

Results show that SBNN outperforms the use of BNN with dropout, both in accuracy and in final sparsity of a large margin, suggesting that, in their current setup, BNNs are not suited for introducing sparsity.

