# OpenReview forum: "Sparse Binary Neural Networks"
_ICLR.cc/2021/Conference — Reject_

### Official Review · AnonReviewer1 · 2020-10-28
**Conceptually sound paper but methodologically very incremental**

**Rating:** 5
**Confidence:** 4

**Review:**

The authors are tackling the problem of sparsifying binary neural networks, i.e. to maximize the number of weights that can be set to zero, without compromising accuracy too much. This is a useful endeavor, both for deploying models on embedded devices where memory might be limited, and to reduce the computation work load at inference time, since less active weights mean less operations to be computed during the forward pass.
The strategy that the authors adopt is to modify binary neural networks like those in the mentioned references of Courbariaux et al. by encoding weights as 0/1, instead of -1/1. In practice, this is done by training on continuous weights between 0 and 1 and adding terms in the cost function that promote the weight values 0 and 1 over intermediate values, and a term that tends to sparsify the weights by pushing them to 0.
The authors test their method on MNIST and CIFAR10 training a 3 layer MLP and a modified version of VGG. In these settings the authors carefully study the effect of the regularization terms that they added and that of the sparsity of the initialization. The main result is that the resulting architectures are remarkable in their level of sparseness: the authors achieve sparseness around 2-3%, with a minimal drop of performance on MNIST but unfortunately a quite substantial one on CIFAR-10 compared to Binary NN (from 88% accuracy to 79%).

While practically valuable as a piece as a demonstration of how tolerant Neural Networks are to reduced precision and sparsification, this paper is methodologically quite incremental, if compared to Binary Net and all the subsequent papers deriving from that one.  The demonstrations are moreover limited to MNIST and CIFAR10, meaning that there aren't any indication of how the method would fare on larger datasets like ImageNet, where the advantages of reduced precision and sparsification are arguably going to be most relevant. In addition, even for a relatively simple dataset like CIFAR10 the proposed method seems to reach good levels of sparsity at a considerable cost in terms of accuracy (more than 9%).
Lastly, previous papers on sparsifying ConvNets have pointed out that the actual advantages in terms of memory, computation and energy consumption would come from reducing the number of filters, rather than the number of active weights.

---

> ### Author Response · Authors · 2020-11-18
> **Some clarifications and a request**
>
> - **Regarding the network topologies used :**
> The choice of the neural network topologies and the datasets chosen are the same used in [1], [2], [3] and [4] to have a fair comparison with other binarization and quantization techniques proposed in the literature. In [1], [2] and [3] also SVHN dataset is used to analyse the effectiveness of the method. In [4] only MNIST is used as dataset.
>
> [1] BinaryConnect: Training Deep Neural Networks with binary weights during propagations (Advances in Neural Information Processing Systems, 2015)
> [2] Binarized neural networks: Training deep neural networks with weights and activations constrained to +1 or -1 (arXiv preprint arXiv, 2016)
> [3] Quantized Neural Networks: Training Neural Networks with Low Precision Weights and Activations (The Journal of Machine Learning Research, 2017)
> [4] Bitwise Neural Networks (Proceedings of the 31st International Conference on Machine Learning JMLR: W&CP, 2015)
>
> - **Regarding missing larger datasets:**
> Thank you for pointing the question about the behaviour of the method on larger datasets. We are currently working to get a new version of the work with the suggested experiment with ImageNet dataset.
>
> - **About the ConvNets future trends:**
> Thank you also for pointing us to the trend in ConvNets to reduce the number of filters. And if you have some references supporting your last sentence, we will be curious to have a look at them.

---

### Official Review · AnonReviewer4 · 2020-10-28
**Introducing sparsity in binary neural networks: high compression rate but limited novelty!**

**Rating:** 5
**Confidence:** 4

**Review:**

Summary of the paper:
This paper presents a sparse neural networks with binary weights. The binary weights are represented using 0s and 1s whereas the effective connections are encoded using the run-length encoder for efficient implementations of neural networks on IoT devices. To quantize the value of weights, a non-negative function is introduced, penalizing values far from two discrete values of 0 and 1. The performance of the obtained sparse binary neural network is measured on MNIST and CIFAR10 datasets when using fully-connected and convolutional neural networks. It was shown that sparsity degree of 98% can be achieved at the cost of an accuracy degradation.

Strength:

-- The paper is easy to read

-- The compression rate is very high

-- Comprehensive simulations for different hyper-parameters

Weakness:

-- Misleading notations

-- Simulation results are obtained from small datasets

-- Significant overlaps with previous works (limited novelty)

Detailed comments:
-- Section 2 under the label “Sparsity”: It is not clear what the authors mean by using {0, 1} for weights instead of {-1, +1}. In conventional methods such as binaryconnet, the value of weights is mapped to two discrete values of -1 and +1 where these discrete values can be represented using a single bit (i.e., 0 for representing -1 and 1 for representing +1) for hardware implementation. Do you mean that your method maps the real value of weights to positive values of 0 and 1? If this is the case, please explain how your binary neural network works with positive weights.

-- Section 3.1: According to Eq. (4), 0 is an representation for -1. If this is the case, what is the intuition behind using the function f2(w) to push values towards -1? On the other hand, if 0 is in fact representing a zero-valued weight, then we have a neural networks with positive weights. Please elaborate.

-- Section 3.2: In the definition of Q, we have mapping from [0, 1] to {0, 1}. However, if you use a = -1 and b = +1 in Eq. (4), then the mapping would be from [-1,+1] to {-1, +1}. Please explain this discrepancy.

-- Section 3: In general, the proposed binarization process is very similar to the work of Courbariaux et al. (2016). Please list the differences.

-- Section 3.3: Initializing a network with sparse connections using a Bernoulli function was introduced in [1]. Please explain the differences.

[1] Sparsely-Connected Neural Networks: Towards Efficient VLSI Implementation of Deep Neural Networks (ICLR 2017)

-- Section 4: I would like to see some experimental results on a bigger dataset such as ImageNet. The quantization of neural networks has been well studied during the past few years. However, they fall short in providing a good accuracy for state-of-the-art networks on ImageNet. I would like to see how the proposed method of this paper stands out against other existing works.

-- Section 4: It would be interesting to see a comparison with ternary neural networks (e.g., [2]) in terms of sparsity degree and accuracy.

[2] Neural Networks with Few Multiplications (ICLR 2016)

-- Section 4: I am also wondering why there is significant gap between the proposed binary network and its full-precision counterpart (which is not included in this paper) given the fact that small datasets are used. Using a high compression rate could be the answer, but it would be interesting to see a curve showing accuracy for different degrees of sparsity (i.e., EC).


--------------------------------------------------------------------------------
Post-rebuttal update: Based on the improvements made during the rebuttal period, I have raised my rating. I believe Fig. 2 now makes the contribution of this paper more clear when compared to the conventional binary network. According to Fig. 2, the proposed method in this paper allows to multiplex and select binary inputs rather than performing XNOR operations, which makes the novelty of this paper rather limited and incremental. Therefore, I still believe this paper stands below the acceptance threshold.

---

> ### Author Response · Authors · 2020-11-18
> **Answer to reviewer’s remarks**
>
> - **Limited Novelty**
> We will try to rephrase our contributions to better highlight them. While we are aware that we build from previous works, our key contribution is the behaviour analysis of binary networks with positive weights to further compress deep neural network models, while showing their tendency to keep sparsity, while faster generalizing when trained with already sparse weight initialization.
>
> - **Detailed comments: Section 2**
> Thanks for pointing this. We understand that this is not fully clear and we will reformulate it in the version currently prepared to express better that we have a network with positive weights (0/1), while having antipodal (-1/+1) activation functions for each neuron.
> - **Section 3.2**
> Activations and weights are quantized differently. We try to reformulate to avoid a misunderstanding. The weights are positive 0/1, while the activation function at each neuron is the sign function, producing as outputs -1/+1.
>
> - **Work similar to ICLR 2017**
> Thanks for pointing us to this work. We will cite them and explain the key differences. Please note that their final result is a ternary weight network instead of a binary weight one {-1,0 for missing connections,1}. Their work focuses on a hardware-friendly Bernoulli function to obtain sparse binary and ternary weight models in hardware, with VLSI technology in mind, while we focus on binary neural networks which can exploit bitwise operations in common processors, with in mind very cheap and memory limited processors, proposing also encoding schemes for the storage of such models. Moreover, they use a fixed mask to remove parameters, while we let the network to build its connections from a starting connection setup.
> - **Section 4**
> We are currently working to get a new version of this work with the suggested experiments. Thanks again for the suggestions.

---

### Official Review · AnonReviewer3 · 2020-11-01
**novelty**

**Rating:** 4
**Confidence:** 4

**Review:**

this paper propose sparse binary neural network for model compression and efficient inference on IoT devices. The paper combines pruning techniques and binary neural network techniques, to train a model that is ternary: {1, 0, -1}. the paper experimented with four regularization functions: triangular, L2-inspired, piece-wise parabola and a 4th degree polynomial. My concern is that, evaluated on MNIST dataset, the accuracy is a bit worse than the BNN, and the implementation section is thin, therefore it's not clear whether the sparsity gain can offset the implementation overhead for decoding. I'm also concerned about the novelty combining pruning techniques with binary neural network techniques.

---

> ### Author Response · Authors · 2020-11-18
> **Some clarifications about the scope of our work and implementation**
>
> - **Scope:**
> We would like to clarify that the proposed model is not ternary, but binary. Weights are binary in {0,1}, while activation functions are binary in {-1,+1}. While ternary models have weights and activations in {-1,0,1}, and they always perform better than binary models.
> - **Implementation:**
> There is not overhead in our method for decoding, if the model is compressed via index encoder. This is because instead of a XNOR and then popcount operation, there is only popcount operation. We thank you for this remark, which we are taking into account in the version we are preparing. We will provide a visual representation of the operations in the rebuttal version.

---

### Official Review · AnonReviewer2 · 2020-11-02
**A clear rejection**

**Rating:** 3
**Confidence:** 5

**Review:**

This paper compresses neural networks via so called Sparse Binary Neural Network designs. The proposed idea is naïve, directly using a slightly modified sign function to quantize network weights into 0 and 1 instead of commonly defined -1 and 1. Experiments on small MNIST and CIFAR-10 datasets with two shallow and old neural networks are provided.

This paper has obvious weaknesses.

--- Limited novelty

The proposed method is naïve. The authors merely replace  binary weights {-1, 1} by {0, 1}, using common quantization tricks for binary neural networks, such as straight-through estimator (STE) and a slightly modified sign function. The authors claim that such a modification can bring significantly improved compression. However, it is problematic, as it will force all quantized network weights to be non-negative, leading to serious accuracy drop. For example, in Table 3, a shallow VGG-like network on CIFAR-10 dataset shows about 10% absolute accuracy drop compared to the binary weight counterpart. Furthermore,  in the optimization, the authors add two non-negative constraints, which makes the training with STE even more challenging. I believe, experiments on large-scale image classification dataset such as ImageNet with modern CNNs will lead to more serious accuracy drops.

Actually, more impressive neural network compression yet with good accuracy can be achieved via combing quantization and pruning. There exist numerous works in this field,  e.g., "Deep compression: Compressing deep neural networks with pruning, trained quantization and huffman coding" in ICLR 2016, "Clip-q: Deep network compression learning by in-parallel pruning-quantization" in CVPR 2018 and "APQ: Joint Search for Network Architecture, Pruning and Quantization Policy" in CVPR 2020. Unfortunately, they are missed by the authors.

--- Poor writing

The paper is poorly written, including introduction (messy), related works (poor), proposed method (tedious) and experiments presentation (weak).

--- Weak experiments

There is no comparison with state of the art CNN compression methods, combining quantization and pruning for improved compression.

The authors only conduct toy experiments, a 3-layer fully connected LeNet on MNIST dataset and a shallow VGG-like network on CIFAR-10 dataset.

Even with toy experiments, results are very weak, showing serious accuracy drop even for a shallow VGG-like network on CIFAR-10 dataset, compared to the binary weight counterpart.

---

> ### Author Response · Authors · 2020-11-18
> **Clarifications and some questions**
>
> - Regarding the use of “shallow and old neural networks”
>
> The choice of the neural network topologies and the datasets chosen are the same used in [1], [2], [3] and [4] to have a fair comparison with other binarization and quantization techniques proposed in the literature. In [1], [2] and [3] also SVHN dataset is used to analyse the effectiveness of the method. In [4] only MNIST is used as dataset.
>
> [1] BinaryConnect: Training Deep Neural Networks with binary weights during propagations (Advances in Neural Information Processing Systems, 2015)
> [2] Binarized neural networks: Training deep neural networks with weights and activations constrained to +1 or -1 (arXiv preprint arXiv, 2016)
> [3] Quantized Neural Networks: Training Neural Networks with Low Precision Weights and Activations (The Journal of Machine Learning Research, 2017)
> [4] Bitwise Neural Networks (Proceedings of the 31st International Conference on Machine Learning JMLR: W&CP, 2015)
>
> - Regarding limited novelty
>
> To counteract the problem of non-negative weight, the activation function used includes {+1,-1}. While in many common training settings of full-precision networks, the weights are real, but the activation function is non-negative (e.g., ReLu activation function is non-negative).
>
> - Regarding non-cited papers
>
> Please note that the first suggested paper (ICLR 2016) is already cited in the submitted version. Among the three suggested citations, this one is the only one to get compressions of around x35 to x49 times, while limited speed-up in CPU when implemented the compressed model. Moreover, this compression can still be insufficient in very cheap sensors.
> Regarding the other two suggestions (CVPR 2018 and 2020), these two works perform quantization a posteriori and not during the training phase, as we do. We will add the two references as examples of such category in the new version we are currently preparing. However, we would like to remark that the CVPR2018 reference has only with a precise model a compression of x51 times, while with other models it compresses no more than x10 to x15 times, which is clearly worse than our results. The CVPR2020 reference mentions a compression aware scheme, but no results about the final compression rate.
>
> - Regarding poor writing
>
> We are in the process of preparing a new version, where we are willing and totally open to restructure the content in the different parts. We would therefore be grateful if more details could be provided. While some other reviewers consider the writing a strength of this work, the provided feedback of this review is not specific on what is considered as poor so, it is difficult for us to establish what is not clear.
>
> - Regarding weak experiments
>
> We do not focus on CNNs compression. We would like to highlight again that some of the works which have been mentioned in this review are less powerful than the proposed scheme.
> We have not used a LeNet on MNIST, but a fully connected multi-layer perceptron topology, the same used in [1], [2], [3] and [4] and used to see the effect of the quantization on fully connected topologies. The shallow VGG-like network is similar to the one used in [1], [2] and [3]. Therefore, we consider that our experimental setup matches that one of state-of-the-art works.
>
> - Regarding accuracy loss
>
> We are aware of the loss and, we are also aware that in general binary neural networks have difficulties with convolutional neural networks. As explained in [5] it is still unclear what kind of network structure is suitable for binarization, so that the information passing through the network can be preserved, even after binarization.
> Our goal is to further enhance the compression capabilities of binary neural networks, which exhibit interesting speed-up properties when run on CPU and low energy consumption, making them very good candidates for very cheap IoT sensors with limited resource battery (E.g., very cheap wireless sensors used to read the digits of sensor meters not connected to the network).
>
> [5] Binary Neural Networks: A Survey (Pattern Recognition, 2020)

---

### Author Response · Authors · 2020-11-24
**New version and missing experiments**

We have uploaded a new version of the paper where we address many of the points raised by the reviewers. Due to time constraints and given the available resources, we were not able to complete experiments on ImageNet.

One reviewer had suggested to include experiments with Ternary Networks. We have found [1] and [2] where the authors focus on enhancing the sparsity of ternary networks, [1] also with MNIST and CIFAR-10, however we were not able to find their source codes for training their methods with the network topologies used in our paper, and the same number of epochs, in order to have a fair comparison with those state-of-the-art methods.

Finally, regarding accuracy at different sparsity levels, the gain relaxing the regularizers is very small, for this reason we think that a future direction can be in extracting these networks from sparse ternary networks, as explained in the conclusion part.

[1] Compressing low precision deep neural networks using sparsity-induced regularization in ternary networks (International Conference on Neural Information Processing, 2017)

[2] Learning sparse & ternary neural networks with entropy-constrained trained ternarization (Proceedings of the IEEE/CVF Conference on Computer Vision and Pattern Recognition Workshops, 2020)

---

### Decision · Program_Chairs · 2021-01-07
**Final Decision**

**Decision:**

Reject

**Comment:**

This paper compresses neural networks via so called Sparse Binary Neural Network designs. All reviewers agree that the paper has limited novelty. Experiments are only performed on small datasets with simple neural networks. However, even with toy experiments, results are very weak. There is no comparison with the SOTA. Numerous related works are missed by the authors. Besides, the paper is poorly written, and there are misleading notations.